# Global constraints within the developmental program of the *Drosophila* wing

**Vasyl Alba[1,2], James E Carthew[1], Richard W Carthew[2,3], Madhav Mani[1,2,3]***

[1]Department of Engineering Sciences and Applied Mathematics, Northwestern University, Evanston, United States; [2]NSF-Simons Center for Quantitative Biology, Northwestern University, Evanston, United States; [3]Department of Molecular Biosciences, Northwestern University, Evanston, United States

**Abstract** Organismal development is a complex process, involving a vast number of molecular constituents interacting on multiple spatio-temporal scales in the formation of intricate body structures. Despite this complexity, development is remarkably reproducible and displays tolerance to both genetic and environmental perturbations. This robustness implies the existence of hidden simplicities in developmental programs. Here, using the *Drosophila* wing as a model system, we develop a new quantitative strategy that enables a robust description of biologically salient phenotypic variation. Analyzing natural phenotypic variation across a highly outbred population and variation generated by weak perturbations in genetic and environmental conditions, we observe a highly constrained set of wing phenotypes. Remarkably, the phenotypic variants can be described by a single integrated mode that corresponds to a non-intuitive combination of structural variations across the wing. This work demonstrates the presence of constraints that funnel environmental inputs and genetic variation into phenotypes stretched along a single axis in morphological space. Our results provide quantitative insights into the nature of robustness in complex forms while yet accommodating the potential for evolutionary variations. Methodologically, we introduce a general strategy for finding such invariances in other developmental contexts.

*For correspondence:
madhav.mani@northwestern.edu

Competing interests: The authors declare that no competing interests exist.

## Introduction

One of the most salient features of organismal development is its reproducibility and tolerance to environmental and genetic perturbations. The robust development of intricate body structures is striking in light of the complexity of the process, involving a vast number of molecular constituents and interactions across multiple spatial and temporal scales. Despite insights into the molecular complexity of an organism's developmental program and the intricacies of the mechano-chemical dynamics involved, we lack a deep understanding of how robustness emerges within a complex developmental program and what its generic quantitative signatures must be (*Visser et al., 2003*). The existence of robustness hints at the existence of global constraints within the development program that ensure that the inevitable fluctuations in underlying mechanisms invariably lead to similar outcomes.

It has been noted that there is an apparent paradox related to robustness (*Wagner, 2008*; *Hansen, 2006*; *Masel and Siegal, 2009*; *Kirschner and Gerhart, 1998*; *Masel and Trotter, 2010*). If some body structure shows evidence of evolving within or between species, it would seem that robustness would antagonize its evolvability. It suggests that there is a conflict between the demands for developmental robustness and the requirement for evolvability. In particular, global constraints on the spectrum of developmental outcomes restrict the nature of varieties that can be selected for by the environment. One explanation to resolve this conflict supposes that the

constraints can be temporarily alleviated, resulting in sudden expression of phenotypic diversity from the underlying genetic diversity (*Waddington, 1953*; *Rutherford and Lindquist, 1998*; *Hermisson and Wagner, 2004*). It is unclear how the constraints are alleviated during the natural evolutionary history of organisms. Another explanation supposes that the constraints allow for many different genotypes to generate similar, though not identical, phenotypes (*Wagner, 2008*) Thus, there are many mutations and combinations of mutations that can elicit similar phenotypic change. These very constraints might ensure that the map from an organism's genotype to phenotype is not idiosyncratic.

To address these issues, we put forward a general mathematical strategy for the direct quantification of the degree and nature of developmental robustness of a wholly integrated body organ. Historically, developmental robustness has been measured by focusing on one or a few discrete traits that are small parts of a much larger body structure (*Waddington, 1953*; *Dun and Fraser, 1958*; *Rendel, 1959*). Moreover, methods to quantify the morphology of large body structures have all relied on an arbitrary fragmentation of the structure into a handful of composite measurable traits, referred to as landmarks (*Kendall, 1989*; *Bookstein, 1993*; *Dryden and Mardia, 2016*; *Abouchar et al., 2014*; *Debat et al., 2009*; *Klingenberg, 2009*; *Klingenberg and Gidaszewski, 2010*; *Pélabon et al., 2006*). Logically then, landmark-based analysis of morphology is potentially missing important features of morphological variation and an incomplete measure of more global properties such as robustness. We propose that a landmark-free approach to align and statistically analyze a large number of individual variants defines the appropriate statistical ensemble to make apparent any global constraints in play within a developmental system.

We have chosen the model system of the wing of *Drosophila melanogaster* to conduct this analysis. We chose the wing as the prototype body structure to study because of the comprehensive understanding of its development, with many genes implicated in wing development (*Lecuit et al., 1996*; *Lecuit and Cohen, 1998*; *Lecuit and Cohen, 1997*; *Blair, 2007*), its complex form and function (*Figure 1a*), and its clear fitness impact for the organism (*Garcia-Bellido and de Celis, 1992*; *de Celis, 2003*; *De Celis and Diaz-Benjumea, 2003*).

Carrying out spatial correlation analysis of large ensembles of aligned individual wings, over many genetic and environmental perturbations, we decompose the data to reveal global features of the developmental process. Remarkably, we find that the outcomes of wing development can be statistically described by a one-dimensional (1D) linear manifold in morphological space that corresponds to a non-intuitive combination of structural variations across the wing. This dominant mode is systematically excited by variants generated by very weak mutations in signaling pathway genes as well as by thermal and dietary environmental perturbations. As such, our work provides direct empirical evidence for the presence of global constraints within the developmental program of the wing, funneling environmental inputs and genetic variation into phenotypes stretched along a single axis in morphological space. Our work is a step towards a quantitative characterization and explanation of the robustness of complex living forms. While the developmental outcomes are globally constrained, there exists the potential for morphological variation along the unconstrained direction. This provides a dissolution of the apparent conflict between robustness and evolvability. Finally, this general strategy used to study the wing formulates an approach for finding such constraints in a broader class of developmental processes.

## Results

### Landmark-free morphometrics

Current approaches to morphological phenotyping are particularly problematic for body structures with complex form and pattern, where spatial information is often arbitrarily discretized into landmarks – anatomical loci that are homologous in all individuals being analyzed (*Figure 1a–c*; *Jones and Mahadevan, 2013*; *Choi and Mahadevan, 2018*). Analysis of phenotypic variation has additional challenges owing to the inability to perform precision alignment of complex two-dimensional (2D) and three-dimensional (3D) shapes, which cannot be achieved through simple schemes that account for rotation, scaling, and shear transformations (*Kendall, 1989*; *Bookstein, 1993*; *Dryden and Mardia, 2016*).

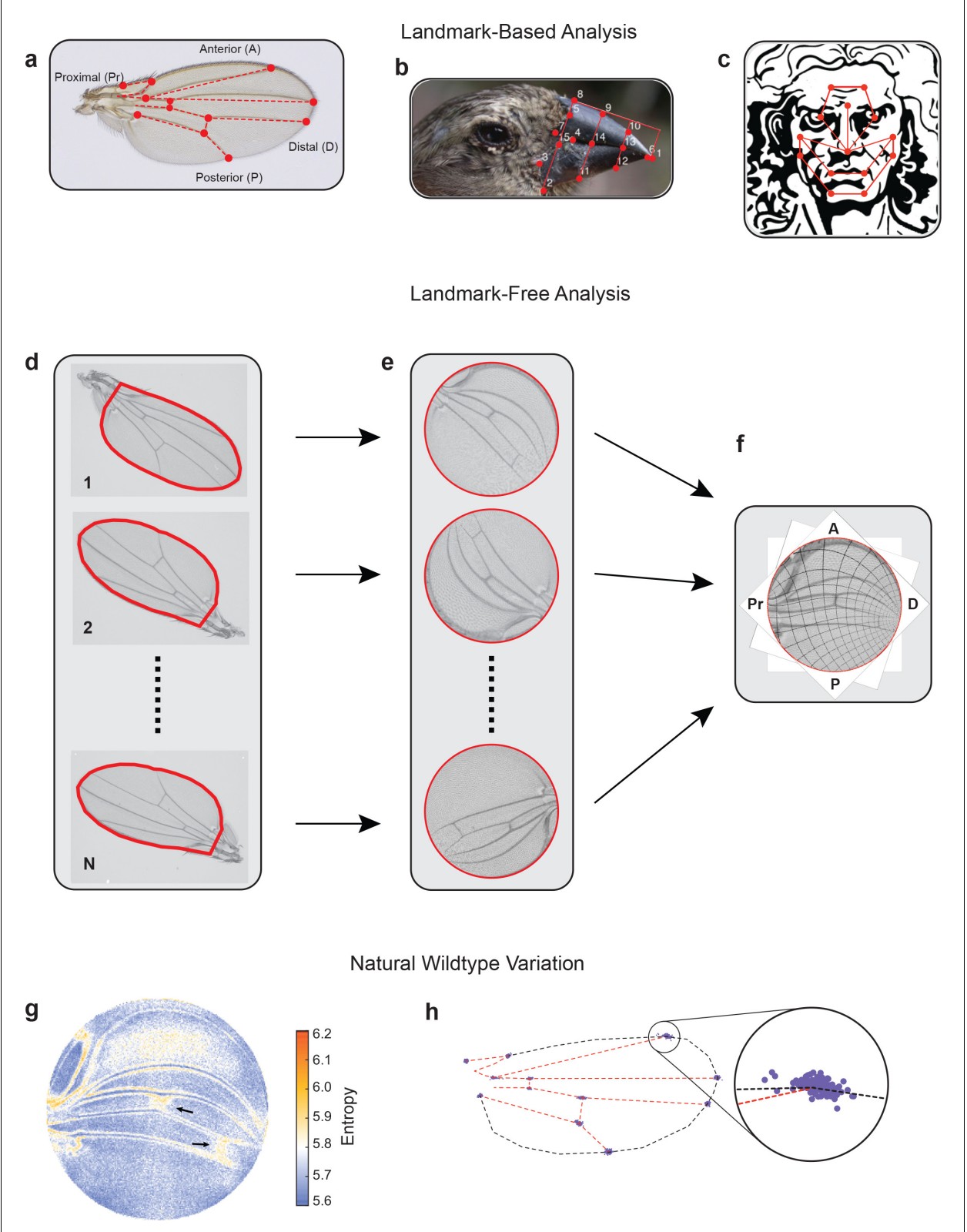

**Figure 1.** Landmark-free morphometrics. (a–c) Standard landmarks (red) for Procrustes analysis of a *Drosophila* wing (a), the beak of a Darwin finch (b), and the face of DaVinci's Vitruvian Man (c). (d–f), The landmark-free method involves boundary identification of wings (d), boundary alignment through conformal mapping to unit discs (e), and bulk alignment of wings through optimization in the space of conformal maps from the disc to itself (f). (g) Pixel entropy of an ensemble of wings from the outbred wildtype population. Enhanced variation of cross-veins is observed (arrows). (h) Procrustes

*Figure 1 continued on next page*

*Figure 1 continued*

analysis of the same ensemble of wings, highlighting variation in the 12 landmark positions. Inset shows the landmark where the L1 and L2 veins intersect.

The online version of this article includes the following figure supplement(s) for figure 1:

**Figure supplement 1.** Procrustes alignment of landmarks.
**Figure supplement 2.** Wing boundary identification.
**Figure supplement 3.** Structure and form of the *Drosophila* wing.
**Figure supplement 4.** Conformal map representation.

To overcome these problems, we have developed an alignment tool that admits a landmark-free method to measure the physical totality of all traits in a body subsystem. Using the wing of *D. melanogaster* as the prototype subsystem, the landmark-free method comprises four steps (*Figure 1d–f*). First, each wing is imaged by transmitted light microscopy at high resolution (*Figure 1—figure supplement 3b*). Second, the boundary of the wing image is accurately detected (*Figure 1—figure supplement 2*; *Berg et al., 2019*). Third, the boundary of the wing image is computationally mapped to the interior of a fixed-sized disc. The mapping of the boundary to the disc relies on an efficient numerical implementation of the Riemann mapping theorem (*Riemann, 1867*; *Ahlfors, 1953*; *Driscoll and Trefethen, 2002*; *Ablowitz et al., 2003*). In particular, the map is conformal, preserving the shape of the image, locally, by preserving angles, while manifestly distorting areas (*Figure 1—figure supplement 4*). This feature of the approach is both a strength and a limitation, downweighting the effects of variation in boundary shape, and upweighting the effects of the pattern of hairs and veins in the interior of the wing (*Figure 1—figure supplement 4j, k*). Fourth, mapped images are globally registered to one another (*Figure 1e, f* and *Figure 1—figure supplement 4l*) since disc-mapped wings vary in their orientation and region of focus (the central point in the disc). These misalignments are spanned by the space of conformal maps of the disc onto itself, which can be optimized to produce a global registration of the images, as shown in *Figure 1f* (further details can be found in Materials and methods). We can thus generate boundary and bulk registered ensembles of wing images, whose variation we can study at single-pixel resolution in a landmark-free manner.

We imaged wings from a highly outbred population of wildtype *D. melanogaster*. This population was founded from 35 inbred wildtype strains collected worldwide, which were then blended together for 2 years to ensure substantial genetic variation. We visualized the quantitative variation in the ensemble of images at the single-pixel resolution, measured as per-pixel information entropy (*Figure 1g*; *Kozachenko and Leonenko, 1987*). Entropy is a measure of the uncertainty, which here is based on variations in image intensity determined on a per-pixel basis (*MacKay and Mac Kay, 2003*). Regions of higher entropy straddle the longitudinal veins, indicating that the positions of these veins along the anterior-posterior (AP) axis vary slightly between individuals; less than a single vein width across the ensemble of wings. However, the proximal region of the L1 vein shows considerable variation, as do some other regions near the hinge. In addition, the cross-veins substantially vary between individuals along the proximal-distal (PD) axis. The intervein region between L1 and L2 veins and the intervein regions near the distal tip also show variation owing to variation in wing hair location. A Procrustes analysis of the same wing ensemble is limited to the variation in landmarks alone, with the intersection of L1 and L2 veins being the most variable (*Figure 1h* and *Figure 1—figure supplement 1c*). An anatomical description of the *Drosophila* wing can be found in *Figure 1—figure supplement 3c, d*.

The wings of male and female *Drosophila* differ in their size, shape, and pattern (*Figure 2—figure supplement 1a, b*), providing a setting to compare the landmark-based and landmark-free approaches to distinguishing two populations. By Procrustes landmark analysis, the intersection of L1 and L2 veins shows the greatest sexual dimorphism, with substantial non-overlap between male and female (*Figure 2a*). The intersection of L1 and L5 veins also shows sexual dimorphism. A very different perspective is revealed using our landmark-free method (*Figure 2b*). The relative entropy of pixel intensities across the two ensembles demonstrates that sexual dimorphism is not restricted to the landmarks. In particular, the proximal and distal regions of the L1 vein are variable between male and female. The L3 vein proximal to the cross-vein also shows considerable variation. Strikingly, the regions showing sexual dimorphism do not spatially overlap with the regions showing variation

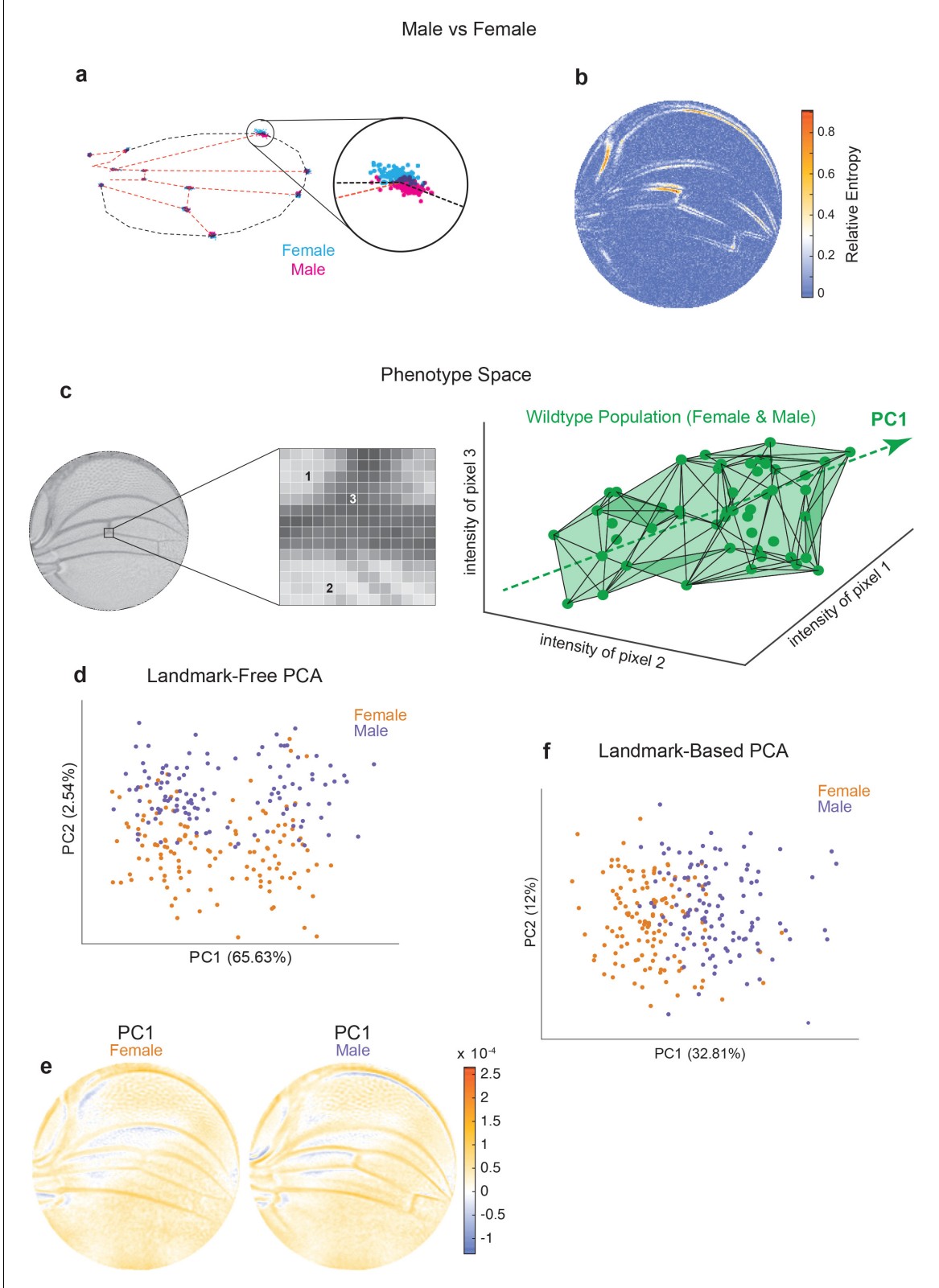

**Figure 2.** Comparison of Procrustes and landmark-free phenotyping. (a, b) Variation between ensembles of male and female wings from the outbred wildtype population. (a) Procrustes analysis with inset showing the landmark where L1 and L2 veins intersect. (b) Per-pixel symmetrized Kullback–Leibler divergence in landmark-free analysis detects variation along veins, undetected by landmark-based analyses. (c) The dimensions of phenotype space comprise individual pixels. Three such pixels are randomly selected here to show their 3D phenotype space. An ensemble of wings are points in this

*Figure 2 continued on next page*

*Figure 2 continued*

space. The direction of largest variation in the space is identified by principal component analysis (PCA) as PC1. (**d**) Landmark-free PCA of the outbred wildtype population reveals a novel dominant direction of variation orthogonal to sex-specific variation. Sex-specific variation aligns with PC2. (**e**) Pixel alignment with PC1 analysis shown in panel (**d**), showing vein position variation aligns with PC1. (**f**) Procrustes-based PCA of the same ensemble analyzed in panel (**d**). PC1 only detects the sex-specific variation.

The online version of this article includes the following figure supplement(s) for figure 2:

**Figure supplement 1.** Accounting for the variation in wings.

**Figure supplement 2.** Mean wing conformal maps from animals raised under different conditions or with different genotypes.

**Figure supplement 3.** Alignment of axis of sexual dimorphism rotates as a function of environmental stress across 4 different conditions.

within each wildtype population. In summary, this comparative analysis demonstrates the remarkable sensitivity of our landmark-free method to make precise measurements of the variational properties of wing form, as well as demonstrating the incomplete picture of variation provided by landmarks alone.

## A dominant mode of natural phenotypic variation

We note that the space in which phenotypes are parameterized via our landmark-free approach is vast. The intensity of each pixel in the space of disc-mapped images is identified with a unique dimension (*Figure 2c*). Therefore, an individual image is a single point in this phenotype space of ~30,000 dimensions (pixels). By contrast, the standard landmark-based parameterization of the wing defines phenotype in 24 dimensions – the *x* and *y* locations of 12 landmarks. By way of illustration, we portray an ensemble of wildtype wings populating a 3D slice of this vast phenotypic space (*Figure 2c*). The dominant mode of variation in the population is indicated as the 'long axis' of the cloud of points. To find the major axes of variation within the 30,000-dimensional space, we performed principal component analysis (PCA) on male and female wings from the outbred stock raised under standard environmental conditions. PC1 and PC2 axes account for 68% of the total variation, with 66% of the variation aligned along PC1 (*Figure 2d*). Strikingly, male and female wings are separated along PC2, suggesting that the dominant mode of variability is not sex-specific. This dominant mode of variability is also not congruent with left-right patterning (*Figure 2—figure supplement 1d*), variation in wing size (*Figure 2—figure supplement 1e*), or variation in imaging conditions across the ensemble (*Figure 2—figure supplement 1f*). We mapped pixels whose variation is most aligned with PC1 and found vein positions are most prominent, particularly in the proximal region (*Figure 2e*). We also performed PCA on the Procrustes landmark data from the wildtype ensemble and observed that sexual dimorphism is the dominant source of detected variation (*Figure 2f*). Strikingly, the dominant mode made manifest in our landmark-free analysis is undetected by a landmark-based method (*Figure 2—figure supplement 1c*), demonstrating that novel, even dominant, modes of variation are rendered detectable by our more holistic analysis.

## Genetic and environmental variation

To investigate the origins of this dominant mode of variation, we performed landmark-free analysis on a published dataset of wing images from the wildtype Samarkand strain and from animals heterozygous for weak mutations in signal transduction genes (*Sonnenschein et al., 2015a*). Specifically, these genes act in the Notch, bone morphogenetic protein (BMP), and EGF receptor (EGFR) pathways, which are required for proper growth and patterning of the developing *Drosophila* wing (*Blair, 2007*). Note that all of the mutations are considered to be recessive to wildtype for wing phenotypes, and thus the heterozygous animals do not exhibit easily discernible differences in wing form from wildtype. Since stronger effect mutations produce phenotypes rarely seen in a natural population, the weak nature of these genetic perturbations is crucial to our investigations into the origins of natural variation. Mutant wing images were subjected to our landmark-free analysis. The high-dimensional phenotype space occupied by each mutant population was compared to the reference wildtype population. This was done by finding the centroid (center of mass) for each population's cloud of points, which constitutes its average phenotype. We then considered the line joining each mutant centroid and the reference wildtype centroid, which represents the difference between the average mutant and wildtype phenotypes. For illustration, a simplified 3D example is shown in

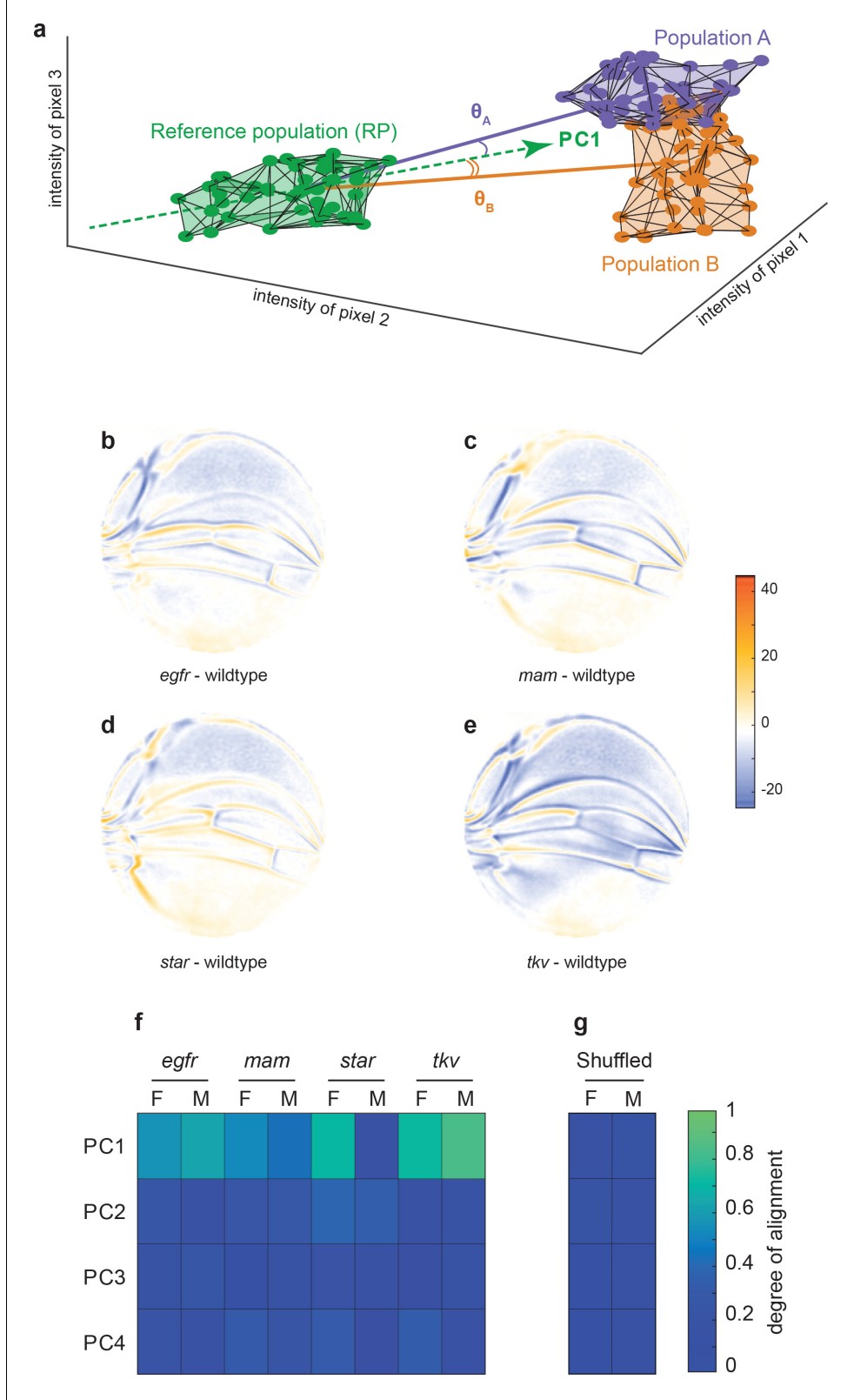

**Figure 3.** Variational analysis of genetic mutants. (**a**) Schematic of hypothetical 3D phenotype space, with each cloud of points representing wings from an ensemble subject to a different treatment or condition. The principal component (PC)1 vector for each cloud of points can be independently calculated. The vectors joining the centroids of any two clouds of points can also be calculated. The angles between the direction of maximal variation (PC1) of the reference population and the directions from the reference to test ensembles are measured (θA and θB). (**b-e**) Per-pixel intensity difference

*Figure 3 continued on next page*

*Figure 3 continued*

between each mutant and the wildtype reference as measured at the center of each point cloud. Negative values are when mutant pixel intensity is smaller than wildtype, and positive values are when mutant pixel intensity is larger. Comparison between wildtype and *egfr* (**b**), *mam* (**c**), *Star* (**d**), and *tkv* (**e**) ensembles. (**f**) The cosine of the angle between the directions of PC1–PC4 of the outbred wildtype population, and the directions of the vectors connecting the wildtype and mutant populations. M and F refer to male and female groups, respectively. (**g**) The cosine of the angle when the wildtype and mutant populations analyzed in panel (**f**) are shuffled but the sexes are not. Shuffling leads to little or no alignment of mutant vectors with any PC. The online version of this article includes the following figure supplement(s) for figure 3:

**Figure supplement 1.** Radon transformation.
**Figure supplement 2.** Eigenvalue spectrum in wings.
**Figure supplement 3.** Bootstrap for genetic ensemble.
**Figure supplement 4.** Statistical significance (p-values) of alignments in genetic analyses relative to a shuffled label null distribution.

*Figure 3a*. The spatial maps of the mutant-wildtype differences in pixel intensities are revealing (*Figure 3b–e*), and they show how our method can detect even subtle effects (see Materials and methods). Mutation of the *Egfr* gene causes L3 and L4 veins to be slightly more distant from one another, whereas mutation of *Star*, which encodes an inhibitor in the EGFR pathway, has the opposite effect. The posterior cross-vein is also affected by these mutants. Both *mastermind (mam)*, a component in the Notch pathway, and *thick veins (tkv)*, a component in the BMP pathway, have common effects in that L2–L4 slightly shift anterior. However, *tkv* shows evidence of subtle vein thickening, which is an attribute of the homozygous mutant phenotype. We conclude that the landmark-free method is sensitive enough to robustly detect phenotypic differences in mutant alleles that are historically considered to be completely recessive to wildtype.

How does the phenotypic variation in the mutants compare with the phenotypic variation in the outbred wildtype population? We measured the alignment between the top four vectors of variation in the outbred population (PC1–PC4), and the vectors connecting Samarkand and mutant centroids (*Figure 3a*). The cosine of the angle $\theta$ between each PC vector and each centroid-centroid vector is reported in *Figure 3f*. There is a high degree of alignment between the dominant mode of variation (PC1) in the outbred population and almost all of the mutants ($\cos(\theta) \to 1$ as $\theta \to 0$). With the exception of the male population of *Star* heterozygotes, the degrees of alignment are far above those expected by chance due to sample size effects or artifacts in the imaging and analysis pipeline (*Figure 3g*). As such, the mutant phenotypes statistically align with a single direction, which is identified as the direction of dominant variation in an outbred wildtype population. We note that the mutants affect distinct signaling pathways, which play significant and separate roles in wing development, thus highlighting the nontrivial nature of the statistical alignment we observe. Taken together, these small-effect genetic perturbations – both the recessive mutations and the variants in the outbred populations – excite variation along a single integrated mode of variation in the wing.

To investigate the nature of developmental variants generated due to perturbations in environment, we raised the outbred wildtype population at different temperatures from the standard 25°C. As previously observed, increasing temperature causes a decrease in wing size (*Figure 2—figure supplement 1g, h*). The differences between the average wing phenotypes at different temperatures were spatially mapped (*Figure 4a, b*). We also raised the outbred population on nutritionally limited food, which decreases wing size (*Figure 2—figure supplement 1i, j*; *Ferreira and Milán, 2015*). The average wing phenotype due to nutrient limitation also reveals robust spatial variation (*Figure 4c*). Remarkably, there is a strong statistical alignment of the temperature and dietary phenotypes with the dominant mode of variation within the outbred population raised under standard environmental conditions (*Figure 4d, e*). Thus, small-effect environmental perturbations are also constrained in their impact on wing phenotype, aligning with the previously identified single mode of variation.

## Geometric analysis of phenotypic variation

Why are some perturbation phenotypes, for example, *Star* mutant male wings, less aligned to the primary mode of natural phenotypic variation than others (*Figures 3f* and *4d*)? Either phenotype is less constrained for some perturbations or some underlying feature of the data was not being considered. Here, we demonstrate that a geometric feature of the data alone explains the observed variations in alignment.

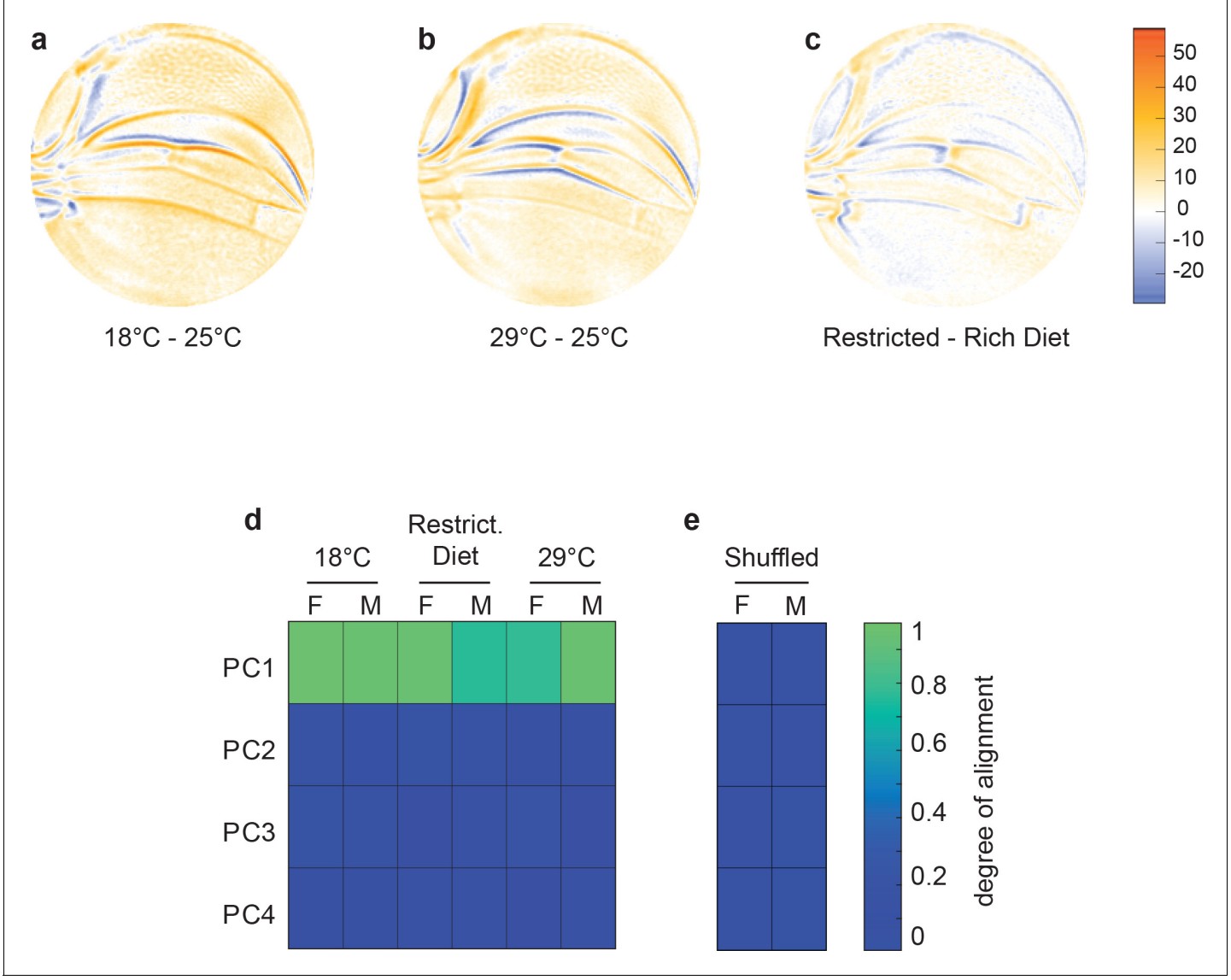

**Figure 4.** Variational analysis of environmental perturbations. (**a–c**) Per-pixel intensity difference between wings from a test condition (described on left) and a reference condition (described on the right). Difference is measured at the center of each point cloud. Negative values are when test pixel intensity is smaller than reference, and positive values are when test pixel intensity is larger. Comparison between wings from 18°C and 25°C (**a**), 29°C and 25°C (**b**), and restricted vs. rich diet (**c**) treatment. (**d**) The cosine of the angle between the directions of principal component (PC)1–PC4 of the outbred wildtype population, and the directions of the vectors connecting the populations raised under different temperature and diet conditions. M and F refer to male and female groups, respectively. (**e**) The cosine of the angle when the treated populations analyzed in panel (**d**) are shuffled but the sexes are not. Shuffling leads to little or no alignment of vectors with any PC.

The online version of this article includes the following figure supplement(s) for figure 4:

**Figure supplement 1.** Kullback–Leibler divergence between left and right wings for each populations of female flies.
**Figure supplement 2.** Eigenvalue spectrum in wings.
**Figure supplement 3.** Bootstrap for environmental ensembles.
**Figure supplement 4.** Statistical significance (p-values) of alignments in environmental analyses relative to a shuffled label null distribution.

---

We consider a situation where wing phenotypes of a population can lie a small distance, $\sigma$, away from the axis of primary variation (PC1) of a reference group (*Figure 5a*). We define $R$ to be the centroid-centroid distance between the two groups, the strength of the phenotypic difference, and $\theta$ to be the angle between the axis of primary variation of the reference group and the vector connecting the centroids of the two groups, a measure of alignment. The three quantities are related through a

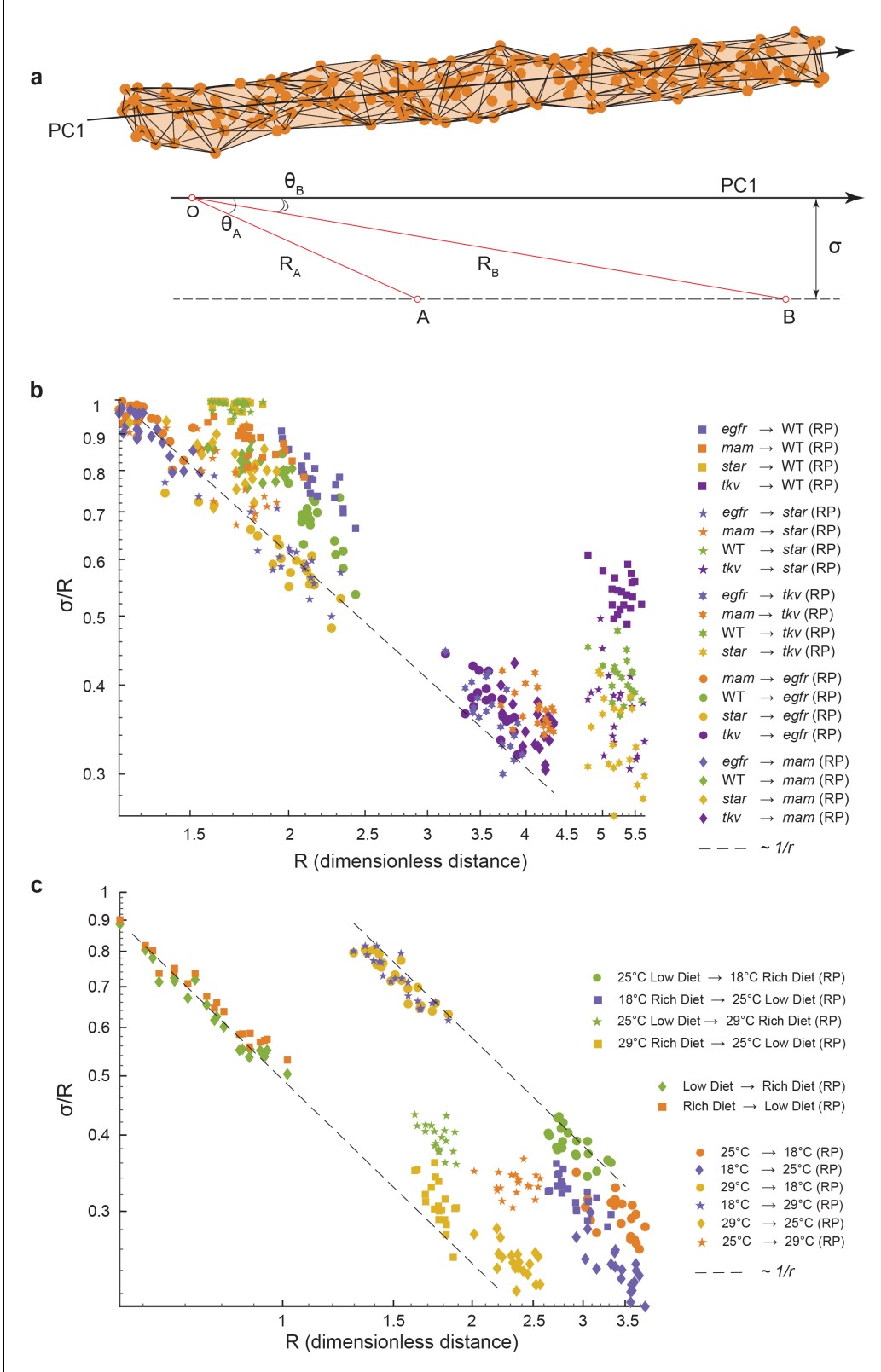

**Figure 5.** Geometric analysis of dominant variational mode. (**a**) Schematic of a data manifold with a single dominant linear direction of variation, embedded in high-dimensional phenotype space. Variation orthogonal to the dominant direction is visualized with a characteristic scale, σ. In the presence of such a data manifold, the angle between the dominant direction of a reference population, principal component (PC)1, and the centroid-to-centroid vector connecting any two populations is predicted to decrease as a function of R, the length of the centroid-to-centroid vector. (**b, c**) The
*Figure 5 continued on next page*

*Figure 5 continued*

angle between the direction of maximal variation (PC1) in a reference population (RP) and the centroid-to-centroid vector connecting the RP and a second population as indicated. This is shown as a function of R, the distance between them. Samples are bootstrapped to indicate the in-sample variance, with each bootstrap represented by a point. Both in the mutant/wildtype (WT) (**b**) and environmental (**c**) ensembles, alignment grows as a function of distance, indicative of a single dominant direction in the data manifold. The dotted lines represent the relationship between σ /R versus R if sigma was a constant.

The online version of this article includes the following figure supplement(s) for figure 5:

**Figure supplement 1.** Geometric analysis of dominant mode.

**Figure supplement 2.** Robustness of manifold geometry to perturbations in image analysis pipeline.

**Figure supplement 3.** Degree of alignment and analysis of global geometry of (**a, b**) log-transformed data and (**c, d**) log-transformed inverse data.

simple trigonometric relation, $\sigma/R = \sqrt{1 - cos^2\theta}$, which mathematically embodies the intuitive fact that if small-effect perturbations can only access a small region of phenotype space perpendicular to the axis of primary variation, then the measure of alignment must increase with the strength of the perturbation. This would mean that $\sigma/R \rightarrow 0$ as $R$ increases (*Figure 5a*). Strikingly, this relation is observed for all of the environmental and mutant perturbations, analyzed by bootstrapped subsampling of the data (*Figure 5b, c*). The results suggest that for all of the genetic and environmental perturbations tested, they generate minor phenotypic deviation orthogonal to the primary mode of natural phenotypic variation. Most variations manifested by the perturbations align with the primary mode of natural variation. For example, the departure of *Star* mutant male wings from alignment with the primary mode of variation (*Figure 3f*) can be explained by the weak phenotypic difference they have, as demonstrated by the short distance between *Star* and wildtype centroids (*Figure 5b*). In summary, we demonstrate that the nature of phenotypic variation caused by diverse weak genetic and environmental perturbations in the fly wing is constrained along a single 1D manifold that aligns with the phenotypic variation seen within every population.

## Discussion

In this study, we present evidence for the presence of strong global constraints within the development program of the *Drosophila* wing. Leveraging a new mathematical strategy to statistically analyze large ensembles of wings across multiple conditions, we demonstrate that natural variants can be described by a 1D linear manifold in morphological space that corresponds to a non-intuitive combination of structural variations that are spatially extended across the wing. Strikingly, the single phenotypic mode is also excited by weak genetic perturbations in distinct developmental pathways, suggesting that its encoding is more global than any particular gene or pathway. As such, our study indicates the presence of strong constraints that limit the spectrum of phenotypic varieties when confronted with small changes in the genetic and environmental conditions of development. The spectrum of investigated phenotypes stand in contrast to the those associated with strong-effect genetic mutations that manifestly probe additional phenotypic dimensions that are absent in natural populations.

We hypothesize two developmental processes that could potentially give rise to such an integrated mode of phenotypic variation. First, thick proveins are refined into narrow veins during pupal development, allowing for the possibility to globally alter the final resting place of veins (*Blair, 2007*). Second, the number, size, and shapes of intervein epidermal cells might vary, and after expansion, separate neighboring veins by different distances (*Kicheva et al., 2007*; *Bollenbach et al., 2008*; *Wartlick et al., 2011*; *Aguilar-Hidalgo et al., 2018*). These effects could be exacerbated in the proximal wing during hinge contraction, which occurs over an extended period of time (*Aigouy et al., 2010*; *Dye et al., 2017*; *Sui et al., 2018*).

We believe that there are three distinguishing characteristics of the approach that facilitated our discovery. First, the precision boundary and bulk alignment of wings permit a more holistic, landmark-free, approach to parameterizing phenotype. Although we apply the approach to one body structure, the *Drosophila* wing, it can be applied to any 2D surface with no holes. This could be a simple 2D shape like the wing or the closed surface of a complex 3D shape such as a tooth, brain, or another body part. Second, transformations that are learned from the data are those that interpolate between different individuals within an outbred *D. melanogaster* population and those subject to

mild genetic and environmental perturbations. This stands in sharp contrast to the more severe phenotypes generated by strong-effect mutations. Third, our approach is statistical, involving the analysis of large ensembles of wings, permitting a quantitative analysis of variance. We propose that it is these three features of our approach that permits the data-driven discovery of global phenotypic constraints in the *Drosophila* wing developmental program and lays the foundations for doing so in other developmental contexts.

Our work presents a general strategy for the quantitative case study of the emergence of robustness and evolvability in complex developmental forms. In the context of the fly wing, we quantify the degree and nature of developmental robustness, which we demonstrate funnels diverse environmental and genetic variation into a 1D, spectrum of phenotypes. Wing development is robust in that the final body structures are statistically constrained to lie along this mode and not elsewhere in morphological space. However, it indicates that the wing is also evolvable because varieties produced along this mode are highly probable. Indeed, our work suggests that the map from genotype to this constrained set of phenotypes is massively degenerate, wherein a vast number and variety of genetic and environmental changes produce similar phenotypic outcomes. As such, our study suggests that global constraints are required to insulate the system from the idiosyncratic and random nature of genes and their mutations. The correspondence between the constrained phenotypic mode identified in this study and fitness remains unclear. The mode might correspond to changes in organismal form that buffer the system from perturbations with little or no impact on fitness. Alternately, the identified mode could correspond to the most broadly conserved aspects of fitness. Disambiguating between these possibilities is central to understand the origins and consequences of such phenotypic constraints. Stepping back, since the spectrum of developmental varieties provide the menu options for the environment to select from, our discovery suggests that closely related species ought to diverge along the phenotypic axis we detect here. If found, an alignment between the statistical properties of a developmental and evolutionary ensemble would lay the groundwork for a quantitative approach to the study of evolutionary-developmental biology.

## Materials and methods

### Fly husbandry

Cornmeal-molasses food was used for standard husbandry. Inbred lines were kept in bottles or vials. The outbred population was kept at 25°C in an 8.8 L population cage. The size of the cage and food availability limited the population size to 1000–2000 adults per generation. Populations were raised in Percival incubators. The temperature of incubators was monitored daily. Fly bottles or vials were monitored for moisture levels. Any condensed moisture on the inside of the container walls was absorbed using Kimwipes. Incubators were monitored for mite infestations using mite traps checked daily.

### Construction of the outbred wildtype population

A collection of 35 wildtype *D. melanogaster* stocks were obtained from the Bloomington Drosophila Stock Center (BDSC). These inbred stocks originated from founder individuals isolated from around the world (*Supplementary file 1*). Fourteen stocks were established in the 20 century from four continents plus the island of Bermuda: one from Africa, three from Asia, one from Bermuda, four from Europe, and five from North America. The other 21 stocks were established in 2003 as isogenic lines from the Drosophila Genetics Resource Panel (DGRP). Founders of the lines were isolated in the area of Raleigh, NC, USA.

Six of the DGRP isogenic stocks were randomly paired with one another, and 20 males from one stock were mated with 20 females from the other stock (*Supplementary file 2*). The offspring from these three crosses were then used in addition to the remaining 29 inbred stocks for a new round of pairwise mating – 16 F0 crosses in total (*Supplementary file 2*). The assignment of stocks to pair was made randomly, as were the assigned sexes used in each paired mating. Again, 20 males from one were mated with 20 females from the other. The offspring from these 16 crosses were used for another round of pairwise mating – eight F1 crosses in total (*Supplementary file 2*). The assignment of stocks to pair was made randomly, as were the sexes used in each mating. For these crosses, 50 males from one were mated with 50 females from the other. The offspring from these eight crosses

were used for an F2 round of pairwise mating – four crosses in total (*Supplementary file 2*). The assignment of stocks to pair was made randomly, as were the sexes used in each mating. For these crosses, 100 males from one were mated with 100 females from the other. The offspring from these four crosses were used for an F3 round of pairwise mating – two crosses in total (*Supplementary file 2*). The assignment of stocks to pair was made randomly, as were the sexes used in each mating. For these crosses, 300 males from one were mated with 300 females from the other. The offspring from these two crosses were used for a final round of mating. For this cross, 800 males from one were mated with 800 females from the other. Mating was performed in a population cage. The population cage was 8.8 L in volume. Offspring from this final cross constituted the outbred stock, the net result of the systematic mixing of 35 distinct populations. The outbred stock was then propagated in population cages for another 50 generations. The purpose of this was for the population to approach equilibrium in allele frequencies.

## Temperature and diet perturbations

Twenty-five mating pairs of outbred adults were mated for 2 days in a bottle, and offspring were raised at a uniform temperature of 18°C, 25°C, or 29°C in Percival incubators. Standard cornmeal-molasses food was used. To compare the effects of rich versus limited diet, 25 mating pairs of the outbred population were mated for 2 days in a bottle, and offspring were raised on food with low yeast content. We used the food recipe described in *Ferreira and Milán, 2015*. The low-yeast food recipe is 1 L food contains 20 g dried bakers' yeast (Bakemark), 100 g sucrose (Sigma), 27 g Bacto Agar (Difco), 3 mL propionic acid (Sigma), and 30 mL 10% (w/v) Tegosept in ethanol (Sigma).

## Wing mounting

Young adults were sorted by sex and stored in 70% (v/v) ethanol for a minimum of 24 hr. Flies were discarded if they had a torn or wrinkled wing. A single fly was dissected in a few drops of 70% ethanol solution on a microscope slide. Wings were removed by clasping the wing joint with a pair of forceps and pulling the wing from the body, using a different pair of forceps to hold the body in place. Both left and right wings were dissected, and the carcass was discarded. Both wings were then placed in a small volume of 70% ethanol solution on the microscope slide ventral side down. A few drops of 70% (v/v) glycerol were added to the wings/ethanol on the slide. If the alula was folded onto the dorsal side of a wing, it was unfolded or removed. A coverslip was placed over the two wings, and any bubbles under the wings were removed by gently pressing down on the coverslip with a pair of blunt forceps. Only the left and right wings from one adult were mounted per slide.

## Microscopic imaging

Imaging was performed using a Zeiss Axioplan microscope. A 4× 0.1 NA Plan Apochromatic objective was used and the Optivar was set to 1.25× magnification. The camera adaptor magnification was 0.5×. This gave a total magnification of 2.5×. The condenser was focused and centered on a wing specimen, and then the stage was moved in order to define shading correction at a neutral point with no slide or wing in the image. Specific shading correction was defined before any images were taken in an imaging session. Each wing had two images taken: one with unfiltered light from the microscope's halogen lamp at 100% power, and one with the green filter in place between the lamp and specimen. The slide remained in the exact same position for each of the two images. After one wing was imaged, its partner wing was then imaged. Wing images were taken so that the anterior side of the wing faced downwards in the image so as to eliminate confusion between right and left wing labeling. If either wing appeared damaged or if dust or debris marked either wing, we rejected that wing and its partner wing. For some female wings, the specimen was too large to be captured in one image. For these, we captured a portion of each wing in one image and then moved the stage to capture the rest of the wing in a separate image. The two images had sufficient overlap that they could be computationally stitched together post-imaging.

Images were captured as CZI files with a Zeiss Axiocam ERc 5s attached to a Dell XPS 8500 computer using ZEN Blue 3.1 imaging software. Each image had dimensions of 2560 × 1920 pixels, and each pixel intensity was recorded in 8-bit RGB mode. Each CZI file was converted to a TIFF file using the Bio-Formats 6.5.1 plugin for Matlab. Exposure time was 0.3 ms for white-lit images and 0.4 ms for green-lit images. Color balance was set to auto. The intensity level was set at 100%. This ensured

that the background was not saturated, that is, background pixel intensity was not maximal. The average background pixel intensity was reproducibly between 210 and 215 for almost all wing samples (*Figure 2—figure supplement 1e*). For those wing samples that were captured using two overlapping images, we used the Photomerge algorithm in Adobe Photoshop CS3 with default settings. Although both white-lit and green-lit images were captured for each wing, we only used the green channel of the green-lit RGB files for further analysis.

We analyzed both left and right wings for each individual. The datasets generated for the following conditions were 18˚C (169 females and 71 males), 25˚C (202 females and 202 males), 29˚C (185 females and 208 males), and 25˚C limited diet (216 females and 219 males).

## Image analysis of genetic mutants

We used a publicly available image database of *D. melanogaster* wings that had been used for landmark analysis by others (*Sonnenschein et al., 2015a*). Briefly, P-element insertion mutants in the *epidermal growth factor receptor (Egfr)*, *mastermind (mam)*, *Star (S)*, and *thick veins (tkv)* genes had been extensively backcrossed into the Samarkand wildtype strain in order to introgress their genetic backgrounds with Samarkand. Each P-element mutant is described as hypomorphic when homozygous, meaning that they are weak loss-of-function mutations. They were crossed to the Samarkand strain to generate heterozygous mutant offspring. These were the wings that had been imaged in addition to fully wildtype Samarkand wings, and all were deposited in a public database (*Sonnenschein and Chari, 2015b*).

We rejected some images due to quality issues that included but were not limited to dust, non-homogeneous illumination, damage to the wings. The remaining images were used for landmark-free analysis: wildtype (214 females and 200 males), *Egfr* (231 females and 236 males), *mam* (212 females and 260 males), *S* (230 females and 222 males), and *tkv* (232 females and 232 males).

## Landmark-based morphometrics

Landmarks are defined as anatomical loci that are unambiguously identifiable and are homologous in all individuals being analyzed. In the case of the *D. melanogaster* wing, 12 points where wing veins intersect have been historically used as landmarks (*Figure 1—figure supplement 1a*). We used the Wings 4 software tool that had been developed specifically to automatically identify the position of the 12 landmarks on each wing. The software fits a spline model of the wing to each picture of the wing. The software and manual can be found in *van der Linde, 2003*.

Once the landmarks were identified for each image, the ensemble of landmarks was aligned using a standard Procrustes method (*contributors, 2020*). In the Procrustes analysis, objects are superimposed by optimal translation, rotation, and uniform scaling of the objects. Both the placement in space and the size of the objects are freely adjusted. The aim is to obtain a similar placement and size by minimizing the Procrustes distance between the objects.

As a hypothetical example, we take a sample triangle and align it with a reference triangle (*Figure 1—figure supplement 1g*). Once the shapes are on top of one another, the sample shape is uniformly scaled to match the area of the reference shape. The last step is to rotate the sample shape to best-align with the reference. These steps are performed until the average distance between the landmarks (vertices of the triangle) of the sample is minimally distant to the landmarks of the reference.

A measure of the goodness of the superimposition is the Procrustes distance:

$$\mathrm{d} = \sqrt{\left(\boldsymbol{x} - \boldsymbol{x_{ref}}\right)^2 + \left(\boldsymbol{y} - \boldsymbol{y_{ref}}\right)^2 + \cdots},$$

where $(x, y, . . .)$ and $(x_{ref}, y_{ref}, . . . )$ are coordinates of the landmarks of the sample image and reference image, respectively. There are 12 landmarks on the *Drosophila* wing, thus there are 24 $x$ and $y$ coordinates that represent each wing. In the case of triangles, $d = \sqrt{r_A^2 + r_B^2 + r_C^2}$ (*Figure 1—figure supplement 1d–g*). If two shapes are exactly the same but have different orientation and position, the Procrustes method will perfectly align them ($d = 0$) (*Figure 1—figure supplement 1h, i*). For the *Drosophila* wing landmarks, we used the Wings 4 software to perform the Procrustes alignment of the landmarks.

Since there are 24 *x* and *y* coordinates that represent each wing, one can think about each wing as a vector in 24−dimensional space. One can perform PCA on these vectors for an ensemble of wings. As a result, one can get directions of the largest variation in landmark positions, but there will not be any information about the positions of the veins or other parts of the wing that have no landmarks. We performed PCA on the landmark-based data using the pca function from the Dimensionality Reduction and Feature Extraction toolbox in Matlab.

## Landmark-free morphometrics

The main concept of the method is to treat every pixel in a wing image as a variable, a trait, with a discrete intensity value. Hence, the higher the resolution of the image, the greater the number of variables. The resolution of wing images we captured are such that there are 100,000 pixels for each wing. However, the pixels are not landmarks in the traditional sense. Traditional landmark-based methods globally align identifiable anatomical loci from homologous objects of different individuals. Then, the variation in the landmark coordinate position is measured for each landmark. Our landmark-free method globally aligns the entire image from homologous objects of different individuals. Then, the variation in intensity for each pixel coordinate is measured.

Homologous body structures from different individuals can have different shapes. This can apply not only to individuals from different species (i.e., forelimb of bat and horse) but also individuals from the same species. This is why the field of morphometrics has reduced the complexity of the continuum of shape to a discretized proxy, that is, landmark-based Procrustes analysis then can estimate shape variation.

Our approach (code is available in *Alba, 2020*) is to use a conformal map to map all homologous body structures onto the same shape, in this case, a disc of fixed size. Although we apply the approach to one body structure, the *Drosophila* wing, it can be applied to any 2D surface with no holes, with or without a boundary. This could be a simple 2D shape like the wing or the closed surface of a complex 3D shape such as a tooth, brain, or another body part.

## Wing boundary identification

Although the background intensity of all images was highly consistent across ensembles (*Figure 2—figure supplement 1e*), there were minor differences, including uneven illumination across the field of view. This variability was most problematic with the wing images from the genetic database (*Sonnenschein and Chari, 2015b*). To correct these issues, we apply contrast-limited adaptive histogram equalization (CLAHE) to our images (*Figure 1—figure supplement 3a*; *Pizer et al., 1987*).

We use an Ilastik segmentation model to identify pixels that belong to the wing bulk, veins, and background (*Figure 1—figure supplement 2a*). The machine learning model is trained on 25 wings and after that is applied to all wings across all ensembles to remove systematic trends owing to variations in segmentation. The output of Ilastik comprises four layers (*Figure 1—figure supplement 2b*). The segmentation output requires further cleaning that is done in a custom-made Matlab script. In particular, we use knowledge about the fact that background is only outside of the wing, and wing bulk is surrounded by veins, to clean up the segmentation (*Figure 1—figure supplement 2c*). Once the segmentation is cleaned, we use a watershed algorithm to produce the boundary of the wing (*Figure 1—figure supplement 2d*). Watershed algorithms find catchment basins in an image, treating intensity as a height of the landscape. Since wings were detached from the body by breaking the hinge region with tweezers, each wing has a slightly different hinge due to experimental manipulation. Therefore, we use two morphologically identifiable landmarks, the humeral break and the alula notch (stars in *Figure 1—figure supplement 2d*). These landmarks are manually selected for each segmented wing image, and a script automatically defines the line between the two points as the proximal boundary. This defines a precise and repeatable way to delete the hinge. Area of each wing is measured as pixel number inside the wing boundary. Distribution of outbred group wing areas is shown in *Figure 2—figure supplement 1g–j*.

## Conformal map conformal mapping

One feature of a conformal map is that it is possible to map any closed region onto another closed region if the shape is approximated as a complex polygon. For example, when we approximate the

boundary of the *Drosophila* wing as a 100-point polygon, we can map the whole wing onto a unit disc. We distribute these points on the wing boundary with a density that is inversely proportional to the local curvature of the wing boundary (*Figure 1—figure supplement 2e*). Therefore, we compute 2D curvature for every point of a segmented wing boundary using the linecurvature2D package (*Kroon, 2020*). Once we have curvature for every point, we compute and plot the cumulative curvature and distribute points equidistantly along the cumulative direction. The cumulative curve is monotonous; therefore, we can find the position of the vertices on the boundary. This approach minimizes the difference between polygonal representation and the real boundary for a given number of points.

Once a polygonal representation is made of a wing boundary, Schwarz–Christoffel mapping generates a map from the interior of the polygon onto a unit disc. The conformal map scales each local region homogeneously, but the scaling constant depends on the position. In the case of the *Drosophila* wing, which is approximately the shape of an ellipse, the bulk of the wing is minimally distorted, while regions far from the I are more distorted. For instance, several peripheral pixels may be mapped onto one pixel of the unit disc (*Figure 1—figure supplement 4k*). Particular distortion depends on the shape of the wing; therefore, if we would like to decouple shape from a pattern, this feature comes in handy.

We use the Schwarz–Christoffel Toolbox for Matlab (*Driscoll, 2002*) to map each segmented wing onto a unit disc that has a diameter of 200 pixels (*Figure 1—figure supplement 4i*). Schwarz–Christoffel conformal mapping requires only the boundary and I-point (*Figure 1—figure supplement 4b*) of the shape. The shape's boundary becomes the circumference and the shape's I-point becomes the origin. This is done individually for all of the segmented wings in an ensemble.

Note that the unit disc is the same area for all segmented wings. A wing boundary is a polygon in a complex space. Therefore, conformal mapping a segmented wing image onto the unit disc depends on the size of the segmented wing image. However, the target (the unit disc) has a fixed size, and so it may be the case that several pixels in the wing image may be mapped onto one pixel in the disc. Consequently, the intensity of the target pixel is an average of the wing image pixels that map to it. For the imaging resolution that was used, this happened most often for peripheral pixels (*Figure 1—figure supplement 4k*).

The Riemann mapping theorem implies that there is a biholomorphic (i.e., a bijective holomorphic mapping whose inverse is also holomorphic) mapping *f* from a non-empty simply connected open subset, $U$, of the complex plane $C$ onto the open unit disc, $D = z \in C : |z| < 1$.

The Schwarz–Christoffel formula is a recipe for a conformal map *f* from the upper half-plane (the canonical domain) $z \in C : Im z > 0$ to the interior of a polygon (the physical domain). The function *f* maps the real axis to the edges of the polygon. If the polygon has interior angles $\alpha_1, \alpha_2, \alpha_3, \ldots$, then this mapping is given by

$$f(z) = f(z_0) + C \int_{z_0}^{z} \prod_{j=1}^{j=1} (w - z_j)^{(\alpha_j/\pi)-1} dw$$

where *C* is a constant, $\alpha_j$ are interior angles at the vertices, and $z_j$ are pre-images of the vertices, that is, dots on the real axis. They obey inequality $z_1 < z_2 < \ldots < z_n = \infty$. For wing data, the unit disc contains 31,416 pixels, which is smaller in number than the ~100,000 pixels in each segmented wing image. The reduction was achieved by reducing the weight of the peripheral pixels, and as a result, the shape of the wing has a lower weight.

## Conformal map alignment

Since wing shapes and orientations are variable, the mapped wings must be globally aligned with one another. To align the mapped images, we use the remaining freedom, which is a rotation of the disc, plus we use the movement of the origin (*Figure 1—figure supplement 4j*). This freedom is an automorphism of the unit disc,

$$g(z) = e^{i\theta} \frac{z - \alpha}{1 - \bar{\alpha}z},$$

where $\theta$ is a rotation angle and $\alpha = g^{-1}(0)$ is a movement of the origin. We perform an

independent parameter sweep of angle and displacement for one mapped wing image to obtain maximal cross-correlation between it and a reference wing disc. This operation is repeated for all of the other mapped wings, aligning each one to the same reference wing disc. We wrote a script that uses the Geometric Transformation Toolbox in Matlab to perform this procedure.

Since we are comparing variation within a group as well as variation between groups, we randomly choose one mapped wing to be the reference for an entire experiment. There is only one reference wing disc for all groups within the experiment varying diet and temperature (G_25C_high_F_139_R). There is only one reference wing for all groups within the experimental data comprising the genetic mutants (G_samw_lei4X_F_110_L).

Left and right wings have a different orientation. We use the function fliplr from Matlab to flip all images of left wings prior to conformal mapping. Once flipped they are treated in the same way as right wings, but we store information about their chirality. To determine if left and right wings are different in shape or pattern, we computed the Kullback–Leibler divergence (for details, see SI.4.14 and *Figure 4—figure supplement 1*) between left and right ensembles for the outbred group and found no difference. We also plot PC1 vs. PC2 for the outbred group and use color labels for chirality (*Figure 2—figure supplement 1d*). We find no correlation between the score for PC1/PC2 and chirality. These analyses indicate that there is no statistical difference between the left and right wings. All subsequent analysis aggregates both left and right wing data.

## Radon transformation

The *Drosophila* wing icomprises a sparse pattern of dark veins and lighter intervein compartments. We anticipate that much of the variation between wings will be detected as differences in vein thickness, position, and their angle relative to the body axes. A Radon transformation of a density function $f(x,y)$ produces a function $Rf(L) = \int_L f(\boldsymbol{x})|d\boldsymbol{x}|$ defined on a space of all lines $L$. For 2D images, a Radon transformation generates sinograms with coordinates of angle and $R$ (*Figure 3—figure supplement 3a, f–k*). Since veins resemble lines, comparing two lines that differ by an angle is more sensitive to Radon-transformed data (*Figure 3—figure supplement 3b–e*). For this reason, we carry out all further analysis of the data with Radon-transformed images of aligned wing discs, obtained using the radon function in Matlab. However, when we need to present images of a wing, we use an inverse transformation iradon in Matlab.

## Pattern variation within a group

Once all images are aligned to a reference on the disc, the remaining variability reflects biological variation. We use 8-bit grayscale images, which means that each pixel has intensity ranging from 0 to 255. Therefore, a wing image with $N$ pixels can be represented as a single vector in discrete $N$-dimensional space, or to be more precise it is a vector in an $N$-dimensional cube ($[0255]^N$). For a group with $M$ number of wings, the group can be represented as a set of $M$ vectors in $N$-dimensional space. Therefore, we can use all the tools from linear algebra.

We can analyze variability within a group since each group of individuals is subject to natural variation. We analyze this variation using PCA, which finds the directions of greatest variation within the group. Each PC is a unit vector in the same $N$-dimensional space as the wing discs comprising a group. We use the pca function from the Dimensionality Reduction and Feature Extraction toolbox for Matlab for all PCA-related computations.

We can also analyze variability within a group using entropy. Each pixel at a fixed position may have different intensities for wings in an ensemble. If the pixel belongs to a region that has almost no variability in intensity, then it will have small entropy, but in the regions where there is pattern variability, the entropy will be larger. As an example, we can consider vein motion, which means that the position of the vein along either AP or PD axes varies between wings. If a pixel at a fixed position has identical intensity values for all aligned images, then that pixel has zero entropy. If a fixed-position pixel is within a wing vein for some but not all aligned images, then it might have larger entropy. In the intervein regions of wing images, the intensity of fixed-position pixels often varies slightly, so their entropy will be small but nonzero. We use the Kozachenko–Leonenko (*Kozachenko and Leonenko, 1987*) entropy estimator for entropy

$$H(k) = log(c_d) + \log(N-1) - \psi(k) + \frac{d}{N}\sum_{i=1}^{N} log\rho_k(i),$$

where $d$ is the dimension of x and $c_d$ is the volume of the $d$-dimensional unit ball $c_d = \pi^{d/2}/\Gamma(d/2+1)$ for Euclidean norm, $\rho_k(i)$ is the distance of the $i$th sample to its $k$th nearest neighbor, and $\psi(x)$ is digamma function.

## Eigenvalue spectra of intra-group variation

For each ensemble of wings, we performed alignment to the reference wing and Radon transformation. For each population, we performed PCA on a single population. As a result, we obtained spectrum of the eigenvalues; see *Figure 3—figure supplement 2* and *Figure 4—figure supplement 2*.

## Analysis of pattern variation between groups: entropy-like approach

When there are two or more groups and we would like to compute the variability between them, we can estimate the Kullback–Leibler divergence, *DKL*, or relative entropy. Let us consider two distributions $P = P(X)$ and $Q = Q(Y)$, the former one is interpreted as a real distribution (postulated prior probability) and the latter one is an assumed distribution that is measured at the experiment. The value of this functional is equal to the amount of information that is not accounted if $Q$ be taken instead of $P$,

$$D_{KL}(P \parallel Q) = \sum_{i=1}^{n} p_i \log\frac{p_i}{q_i}.$$

We use the two-nearest-neighbor non-parametric Kozachenko–Leonenko estimator (*Wang et al., 2009*) for the Kullback–Leibler divergence,

$$D_{KL}(P \parallel Q) = \frac{d}{N}\sum_{i=1}^{N}\log\frac{\nu_k(i)}{\rho_k(i)} + \log\frac{M}{N-1},$$

where $\rho_k(i)$ is the distance from $x_i$ to its $k$th nearest neighbor in $\{X_j\}_{j\neq i}$, and $\nu_k(i)$ is the distance from $x_i$ to its $k$th nearest neighbor in $\{Y_j\}$.

## Left-right patterning

We present Kullback–Leibler divergence between left and right wings for each population of female flies. For comparison, we also present Kullback–Leibler divergence between two randomly assigned groups that have the same number of wings as left and right groups correspondingly. We conclude that there is no significant difference between left and right wings as between groups and that special noise level is higher for the groups that have smaller number of flies.

## Analysis of pattern variation between groups: centroid-based analysis

Another method to compare two or more groups is by centroid-based analysis. For one group containing Z wings, the group can be represented as a set of Z vectors in *N*-dimensional space. The center of mass or centroid of the set can then be computed. The centroid can be thought of as the mean wing of the group (*Figure 2—figure supplement 2a, b*). Since we have a large number of wings for each group (*Z* > 100), the mean wing is a statistically robust representation of the group (also we used bootstrap to check robustness, see next section). We perform all manipulation for Radon images of the wings and transform everything back with its inverse transformation.

We compute the mean wing for each group of wings, and then we can calculate the pairwise difference between mean wings of different groups. A pairwise difference is a vector that shows how one mean wing transforms into another mean wing. For example, when we say that there is a direction from mean wing A to mean wing B we mean that there is a vector, Δ, that generates mean wing B when added pixel-wise to mean wing A: B = Δ + A. If we normalize this vector, n = Δ/|Δ|, we get a unit vector that points from the reference group to another group (i.e., wildtype group to genetic mutant group). We can compare the direction of a unit vector with the direction of any PC vector

from PCA of a given group. This is done by computing a scalar product between the unit vector and the PC vector:

$$\cos\theta = (\boldsymbol{n}, \boldsymbol{PC}).$$

## Bootstrapping and shuffling

We have to check that the findings of the analysis are statistically robust. This requires that the populations are well-sampled, that is, we have enough images so that the in-sample variance of the mean wing is small, and as a result that measured direction between populations is stable and close to the ideal direction that would be achieved if we had an infinite number of wings. For this approach, we use the bootstrap method that requires that we randomly draw subsets of wings that have a size of 50% of the ensemble. We repeat this procedure for each ensemble individually, so we do not mix wings that correspond to different populations. For pairs of such samples, we compute mean wings and the directional vector between them. We found that the numerical value of the $\cos\theta$ is stable, and the difference between sampled in this way $\cos\theta$ and value that is computed for whole populations is extremely low (few percent of the corresponding values), suggesting that it is well estimated.

The second important check is to confirm that the trends we observe are biological in origin and not artifacts of our analysis. We perform a shuffle test. For this check, we construct new populations from images drawn randomly from the entire ensemble, which means that we mix wings from different populations (difference environmental and genetic conditions). We find that for these shuffled populations, a variation of the $cos\theta$ is relatively large, and the mean value is only a few percent of the unshuffled data. This suggests that the observed phenomenon depends on biological labels of the wings rather than artifacts introduced by the method and analysis itself.

## 1D structure of the observed data

While the data is embedded in high-dimensional space, it does not imply that the structure of the data is high dimensional as well. The simplest structure of the data that we may have is a 1D cylinder, where the data is distributed along a line with noise in directions orthogonal to it. The simple structure of the data would hint at a simplifying principle that could be modeled and more deeply understood.

The angle between the **n** and PC vectors resides in an *N*-dimensional space. For a 1D cylinder, one can compute a dependence of $\cos\theta$ as a function of *R*, the distance between points. If the distance between the two mean wings is *R*, then

$$cos\theta = \sqrt{1 - \left(\frac{\sigma}{R}\right)^2} \Longrightarrow \frac{\sigma}{R} = \sqrt{1 - cos^2\theta}.$$

Thus, we get that $\sqrt{1 - cos^2\theta} \overset{R\to\infty}{\to} 0 \, \text{as} \, \sigma/\text{R}$, which suggests the presence of a long direction in the data that dominates all other directions of variation. It further suggests that where $cos\theta \sim R^{-1}$, then the noise in the directions orthogonal to the cylinder is constant along it. The simplest explanation for a different exponent would be that noise level increases with distance, and the structure is a cone rather than a cylinder. We experimentally measure the value of angles θ for pairs of bootstrapped centroids, computing $\sqrt{1 - cos^2\theta}$, and we found that it qualitatively displays a decay as a power of $R$: $\sqrt{1 - cos^2\theta} \overset{R\to\infty}{\to} 0 \, \text{as} \, \frac{\sigma}{R^\alpha}$. Values of α from fitting to an experimental data are reported in *Figure 5—figure supplement 1*.

## Robustness of results to errors in image analysis

We performed three artificial perturbations of the system in order to check the robustness of the method (*Figure 5—figure supplement 2e–h*). The first one is a random shift of the origin that was obtained by the maximization of the cross-correlation between individual image and reference image (*Figure 5—figure supplement 2f*). We randomly shifted origin by a random value from −3 to 3 pixels in each direction. Also, we randomly shifted alula notch and humeral break by a random value from −3 to 3 pixels in each direction. Once these points shifted, new boundary is identified as presented in *Figure 5—figure supplement 2g*. The final perturbation was perturbation of the boundary segmentation. Each pixel of the initial boundary that was obtained after refinement of the segmentation part of the pipeline is shifted by random number from −1 to 1 (*Figure 5—figure*

*supplement 2h*). After these perturbations, the remaining pipeline is applied. We notice that despite small numerical differences of the analysis for the perturbation, quantitative conclusions remain the same.

## Robustness of results to log-transformed pixel data

Motion of a vein across a database corresponds to a significant variation in intensity in limited zones of the image. We performed the analysis on log-transformed data and log-transform of the inverse of data (where veins are bright as opposed to dark structures) to investigate the significance of the representation. We confirmed that all the statistical trends are maintained with intensity transformed data. We thank the referee for this and other recommendations through which we believe we have provided further evidence towards the accuracy and robustness of the computational approach/ pipeline.

## Statistical significance

We were exploring previously unknown phenomena, and therefore it is a priori unknown as to what is a good observable, let alone what a sufficient sample size ought to be to help parameterize it. In light of this, our study has generated a large dataset that permits a careful analysis of statistical robustness and significance. As a rule, we leverage non-parametric approaches to assessing significance, including bootstrap with replacement and resampling from marginal distributions (also known as shuffling). We pursue a non-parametric approach to avoid any assumptions regarding the generative process of a manifestly very complex phenomena.

This section of the SI puts forward our reasoning for our confidence in our conclusions. In particular, we confirm that our sample size is sufficient to generate robust estimation of inferred parameters that we use in our study.

### Parameters of interest

There are fundamentally two quantities that form the basis of our analyses of intra- and inter-population variation. Those being the first and second moments of each genetic and environmental population in the phenotype space that we define. It must be stated that these two quantities are provably the most statistically robust features of an empirical data distribution. Nonlinear analyses that are nowadays commonplace in even the most standard dimensionality reduction algorithms are more susceptible to noise. These two quantities go into helping a spectral, or principal component, analysis of the distributions. The central result of the paper is the striking dominance and importance of the first principal component (eigenvector). It must again be stated that this direction is the most statistically robust direction by construction. The only other parameter we infer is the mean phenotype of populations, which is already a quantity that goes into the construction of PCA.

### PCA

To demonstrate the significance of the PCs that we further analyze and base our conclusions on, we adopt a marginal resampling approach. The explicit advantage of this approach over a Marcenko Pasteur (MP) approach is that the closed-form analytic solutions for the eigenvalue distributions are based on an assumption of Gaussianity of the data. To avoid this parametric assumption, which is rarely true for real-world data, we adopt a marginal resampling approach. This approach gives a spectrum of eigenvalues that could be generated by the nature of the marginal distributions alone, and not due to any covariance in the data. For all our populations, we got at least 70 significant eigenvalues.

### Degree of alignments

The central feature of our analysis was to estimate the degree of alignment between the (1) direction of variation between the means of two populations and (2) the natural variation of the individual populations. (1) Is the phenotype associated with the difference, say, for example, the phenotypic difference between a genetic mutant set of wings and the WT? The mean values of the degree of alignments are presented in the main manuscript in *Figures 3f* and *4d* for the genetic and environmental analyses, respectively. Complete distributions of these alignments, achieved through

bootstrapping, are presented here in *Figure 4—figure supplement 3* and *Figure 3—figure supplement 3*, for the environmental and genetic populations, respectively.

To assess the significance of these measurements of alignment, you must compare what is measured to a null distributions. Again, in the non-parametric approach embodied in our work, the null distribution is constructed from the data itself. In particular, we compare the measured directions to directions that would emerge by chance were the labels of the different populations of wings to be shuffled. Any alignment between inter-population directions of change and intra-population directions of variance in the shuffled label data is purely due to chance or an artifact of our pipeline. *Figure 4—figure supplement 4* and *Figure 3—figure supplement 4* present the p-values achieved through these analyses in the environmental and genetic populations. In all instances, other than the male star population, the alignments are significant. The lack of significance of the male star population is explained in our manuscript as being a statistical consequence of its very weak effect on phenotype.

### Sexual dimorphism vs. the long axis of the data

We include a more detailed description of the relative orientations of the long axis in the data and the axis corresponding to sexual dimorphism in the high-quality images of environmental variation. We observe that within stressed populations of flies, at low diet and high temperatures close to their physiological limit (29°C), the sexual dimorphism axis rotates from being orthogonal to the long axis in the data to parallel; see *Figure 2—figure supplement 3*. This conditional GxE interaction (G being sex-specific genes) provides further evidence that what we are observing is biological in origin, rather than an artifact of our pipeline. It is also consistent with our observation that alignment of mutational perturbations in signaling genes with the long axis in the data is sometimes dependent on the sex of the flies, depending on the mutation (*Figure 5*).

## Acknowledgements

We thank Seppe Kuehn for fruitful conversations. This work was supported in part by NSF grant DMS-1547394 (JEC), NSF (1764421, MM and RWC), and the Simons Foundation (597491, MM and RWC). MM is a Simons Foundation Investigator.

## Additional information

### Funding

| Funder | Grant reference number | Author |
| --- | --- | --- |
| National Science Foundation | 1547394 | James E Carthew |
| National Science Foundation | 1764421 | Richard W Carthew<br>Madhav Mani |
| Simons Foundation | 597491 | Vasyl Alba<br>Richard W Carthew<br>Madhav Mani |
| Simons Foundation | Investigator - MMLS | Madhav Mani |

The funders had no role in study design, data collection and interpretation, or the decision to submit the work for publication.

### Author contributions

Vasyl Alba, Conceptualization, Data curation, Software, Formal analysis, Validation, Investigation, Visualization, Methodology, Writing - original draft; James E Carthew, Data curation, Investigation; Richard W Carthew, Conceptualization, Resources, Data curation, Supervision, Funding acquisition, Investigation, Visualization, Methodology, Project administration, Writing - review and editing; Madhav Mani, Conceptualization, Resources, Data curation, Software, Formal analysis, Supervision, Funding acquisition, Validation, Investigation, Visualization, Methodology, Writing - original draft, Project administration, Writing - review and editing

## Author ORCIDs

Richard W Carthew ![ORCID] http://orcid.org/0000-0003-0343-0156
Madhav Mani ![ORCID] https://orcid.org/0000-0002-5812-4167

## Decision letter and Author response

Decision letter https://doi.org/10.7554/eLife.66750.sa1
Author response https://doi.org/10.7554/eLife.66750.sa2

## Additional files

### Supplementary files

- Supplementary file 1. Description of inbred stock lines.
- Supplementary file 2. Crossing scheme.
- Transparent reporting form

### Data availability

Data will be made publicly available at Mani, Madhav, 2021, Imaging data from "Global Constraints within the Developmental Program of the *Drosophila* Wing", https://doi.org/10.7910/DVN/UFGJFB, Harvard Dataverse, V1.

The following dataset was generated:

| Author(s) | Year | Dataset title | Dataset URL | Database and Identifier |
|---|---|---|---|---|
| Alba V, Carthew JE, Carthew RW, Mani M | 2021 | Imaging data from "Global Constraints within the Developmental Program of the *Drosophila* Wing" | https://doi.org/10.7910/DVN/UFGJFB | Harvard Dataverse, 10.7910/DVN/UFGJFB |

The following previously published dataset was used:

| Author(s) | Year | Dataset title | Dataset URL | Database and Identifier |
|---|---|---|---|---|
| Sonnenschein A, Zee DV, Pitchers WR, Chari S, Dworkin | 2015 | Supporting material and data for "An Image Database of *Drosophila melanogaster* Wings for Phenomic and Biometric analysis" | http://dx.doi.org/10.5524/100141 | GigaScience Database, 10.5524/100141 |

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
