## [Decision Letter]

**Acceptance summary:**

This paper addresses the interesting problem of understanding how complex metazoan forms can be so robust and tolerant to perturbation. The authors have created a new landmark-free approach to morphological analysis of complex developmental structures (here, the fly wing) and use it to describe variation across individuals in a population and in response to weak genetic and environmental perturbations. Their finding that morphological variation amongst individuals follows along a low-dimensional, but spatially non-intuitive, mode is a fundamental result. The landmark-free analysis should be broadly applicable to other structures and developmental systems.

**Decision letter after peer review:**

Thank you for submitting your article "Global Constraints within the Developmental Program of the *Drosophila* Wing" for consideration by *eLife*. Your article has been reviewed by 2 peer reviewers, and the evaluation has been overseen by a Reviewing Editor and Naama Barkai as the Senior Editor. The reviewers have opted to remain anonymous.

Essential Revisions:

1) It's not entirely clear what is the 'true' answer when comparing phenotypes, and whether the dominant principle component in the data sets could just reflect the mapping method and choices made to delineate the wing. Can the authors test for this by intentionally augmenting wing shape and reference points and ask how these map to the disk? Please see reviewer2 point #1, #2, and #4 for specific recommendations.

2) The data support the notion that there is a low-dimensional character to the magnitude of variation amongst individuals in terms of developmental phenotype. It would be important to include a discussion addressing how is the size of such variations is related to fitness of individuals of a species. Could it be that weaker modes that carry less of the overall variation still are under strong selective pressure in certain environments and indeed, are strongly relevant for organismal fitness? Or is the argument that the global mode simply defines the most broadly conserved aspects of fitness? If the latter, then it is interesting to understand how much of the wing developmental program is captured in the top mode of variation.

3) The key technical feature of this work lies in the mapping of images of wing morphology to a fixed-sized disc and alignment over the ensemble of images. Could the authors discuss whether such a transform is guaranteed for any sort of structures during development? This may help readers understand how general the method developed here might be for enabling similar studies in other developmental systems.

4) Please address questions regarding technical aspects of the method – see reviewer2 point #3, #7.

*Reviewer #1:*

This paper addresses the interesting problem of understanding how the seemingly complex and even baroque developmental processes of metazoan forms can be so robust and tolerant to perturbation. The essence of the idea is to propose that there are hidden simplicities that are not obvious but that can be discovered in a data-driven manner using statistical reasoning. With this premise, the authors claim (1) they have created a new landmark-free approach to morphological analysis of complex developmental structures (here, the fly wing) in which ensembles of images are processed to identify the boundary of the structure, then mapped onto a disc while maintaining local geometry, and then aligned, (2) that this stack of aligned structures can be then statistically analyzed for pixel-by-pixel positional entropy, providing a high-dimensional feature space within which each image is embedded, (3) that the landmark-free analysis is superior to existing low-dimensional landmark-dependent approaches, (4) that eigenvalue decomposition of the ensemble shows that most of the variation amongst individuals is accounted for in just the top mode, (5) that the top modes of variation are different from what is learned by the traditional landmark-dependent approach, (6) that subtle genetic variations due to mutations or natural outbred variations project onto the top mode of this analysis, and (7) that significant perturbations due to temperature during development and or dietary fluctional largely excites variations along the same top mode. From this, the authors propose (1) that there are global, low-dimensional constraints within the development of the fly wing such that random, genetic, and environmental perturbations are largely forced to cause variations along a single mode of variation, and (2) that this mode is not obvious in traditional ways of analyzing morphological data.

In general, this is a very interesting and topical body of work. The question is important, and the specific findings seem well supported by the data presented. The finding that morphological variation amongst individuals follows along a low-dimensional (but spatially non-intuitive) mode is a fundamental result, especially if the approach of landmark-free analysis inspires other developmental biologists to check and verify the claims in many other systems. The finding that genetic and environmental variations also follow along the same global mode is a potentially deep statement about the relative simplicity underlying the vast apparent complexity of molecular mechanisms and more naively observed phenotypes associated with development. If the results of this work can be extended, it could represent an approach for a more general and simpler description of developmental processes and a route to better characterize the underlying mechanisms.

A couple specific points that the authors may wish to consider:

(1) The data support the notion that there is a low-dimensional character to the magnitude of variation amongst individuals in terms of developmental phenotype. But how is the size of such variations related to fitness of individuals of a species? Could it be that weaker modes that carry less of the overall variation still are under strong selective pressure in certain environments and indeed, are strongly relevant for organismal fitness? Or is the argument that the global mode simply defines the most broadly conserved aspects of fitness? If the latter, then it is interesting to understand how much of the wing developmental program is captured in the top mode of variation.

(2) The key technical feature of this work lies in the mapping of images of wing morphology to a fixed-sized disc and alignment over the ensemble of images. Perhaps the authors could indicate whether such a transform is guaranteed for any sort of structures during development. This may help readers understand how general the method developed here might be for enabling similar studies in other developmental systems.

*Reviewer #2:*

The authors implement a novel method for quantifying phenotypic variation in flat two dimensional structures that consists in mapping the entire image into a standard shape (a disk) by an angle preserving transformation. It eliminates the need for preselecting fiducial marks and projecting onto the space of their relative positions. They analyze the *Drosophila* wing, which is nicely two dimensional, and well-studied by other means, thus a good benchmark. The authors find that variation within a population and in response to various weak genetic mutations predominately fall along the same principle component.

It's not entirely clear what is the 'true' answer when comparing phenotypes, and whether the dominant principle component in the data sets could just reflect the mapping method and choices made to delineate the wing. Nevertheless, the authors new method is provocative and deserves to reach a wide audience, who can try it on other data sets.

1. It's difficult to design a data set in this area where the 'ground truth' is known, so that the degree an algorithm discovers that truth is a measure of its quality. The worry then is the authors analysis inserts some variability in a particular way and the primary principle component just 'discovers', the variability that was inserted. I do not know how to definitively test for this, but to check for the obvious things, can the authors:

i. Intentionally augment the outer boundary of the wing, and analyze as in Figure 5. Is R >> σ. What if some ripple was added to the boundary, what would that look like mapped to the disk?

ii. The authors need to rely on a reference feature to map to the origin of the disk. Assume the position of that was randomized by a few pixels, how would that display in Figure 5?

iii. Same set of questions for the way the authors separate the wing from the hinge

2. In Figure 2d most of the variation is in common for male-females under the landmark free methods while with landmarks the largest PCA component discovers sex. The authors claim this is a vindication of their method, but could it in fact be a problem (as suggested in 1.) Also the data in 2d seems to be bipartite, any explanation?

3. Can the authors take their conformal mapped and registered images, and extract the coordinates of the Procrustes reference points. What is the relation of the principle components based on the Procrustes variables versus their pixel based measures. It's not entirely clear to me what feature their dominant PC picks up. If I shift a vein normal to itself a lot of pixels change, but the movement is less consequential viewed as an operation on a vein. Thus as usual an information theory measure can be misleading.

4. A key finding is that genetic perturbations fall along the same PC as does intrapopulation variation. In the case of the gene tkv the authors remark that their analysis picks up the same feature displacement in the weak heterozygote as in the homozygote. Is the same true for the other mutants they examine, in particular N. The N mutants are readily visible, if they are processed as in Figure 5b,c would they fall along the primary PC? If the authors consider it too onerous to measure on new flies, can they just by photoshop impose a comparable mutation on a few of their current images and process?

5. With their environmental perturbations the authors could ascertain when in development the wings are sensitive to the environment, ie does a molt reset the phenotype of the wing?

6. The ultimate test of the unique principle component, is to measure the wings on a closely related species of *Drosophila* (as the authors remark at the end of their discussion). Would this be a big increment in work over what has already been done? Just 10's of individuals from a stock collection of D. simulans, D. schellia or whatever is easiest. Some of these species make sterile hybrids with *D. melanogaster*, so it would be interesting to know how close the wings are.

7. Some technical remarks about the mapping to the disk

i. Is there a reflection among the rotations to align the right and left wings?

ii. Is it obvious one should do PCA directly on the pixel variables, (ie not log transformed for instance). Not many pixels change, but those that do may change their gray scale by a factor of 2-10 (depending on how you are treating background). Does this matter?

The paper is clearly written and should be published after the above questions and those from other referees are addressed. If it's wrong, it's at least interesting.

---

## [Author Response]

Reviewer #1:A couple specific points that the authors may wish to consider:(1) The data support the notion that there is a low-dimensional character to the magnitude of variation amongst individuals in terms of developmental phenotype. But how is the size of such variations related to fitness of individuals of a species? Could it be that weaker modes that carry less of the overall variation still are under strong selective pressure in certain environments and indeed, are strongly relevant for organismal fitness? Or is the argument that the global mode simply defines the most broadly conserved aspects of fitness? If the latter, then it is interesting to understand how much of the wing developmental program is captured in the top mode of variation.

We thank the reviewer for raising these most fundamental issues. First, we have included additional figures in a new section of the SI where the eigenvalue spectrum of each ensemble is presented. Indeed, the first mode carries significantly more variation than the remainder, combined. Second, as the reviewer no doubt recognizes, we have no empirical approach to quantitatively estimate the fitness of wings and/or individuals. That said, the relation between form and fitness is central to what we conclude from our observations. We have now included an additional discussion of the two possible scenarios that the reviewer points out in our Discussion section of the manuscript, which we recount here:

“The correspondence between the constrained phenotypic mode identified in this study and fitness remains unclear. The mode could correspond to changes in organismal form that buffer the system from perturbations with little or no impact on fitness. Alternately, the identified mode could correspond to the most broadly conserved aspects of fitness. Disambiguating between these possibilities is central to understand the origins and consequences of such phenotypic constraints.”(2) The key technical feature of this work lies in the mapping of images of wing morphology to a fixed-sized disc and alignment over the ensemble of images. Perhaps the authors could indicate whether such a transform is guaranteed for any sort of structures during development. This may help readers understand how general the method developed here might be for enabling similar studies in other developmental systems.

We thank the reviewer for pointing out this gap in our discussion of the method. As the

reviewer is no doubt aware, the mathematical foundation of the method outlined in our

study is the Riemann mapping theorem, which holds for a smooth two-dimensional

surface with no holes. We now have a more detailed statement in the manuscript describing this, which we recount here:

“Our approach (code is available at (48)) is to use a conformal map to map all homologous body structures onto the same shape, in this case, a disc of fixed size. Although we apply the approach to one body structure, the *Drosophila* wing, it can be applied to any two-dimensional surface with no holes, with or without a boudnary. This could be a simple 2D shape like the wing or the closed surface of a complex 3D shape such as a tooth, brain, or another body part.”

Reviewer #2:The authors implement a novel method for quantifying phenotypic variation in flat two dimensional structures that consists in mapping the entire image into a standard shape (a disk) by an angle preserving transformation. It eliminates the need for preselecting fiducial marks and projecting onto the space of their relative positions. They analyze the Drosophila wing, which is nicely two dimensional, and well-studied by other means, thus a good benchmark. The authors find that variation within a population and in response to various weak genetic mutations predominately fall along the same principle component.It's not entirely clear what is the 'true' answer when comparing phenotypes, and whether the dominant principle component in the data sets could just reflect the mapping method and choices made to delineate the wing. Nevertheless, the authors new method is provocative and deserves to reach a wide audience, who can try it on other data sets.1. It's difficult to design a data set in this area where the 'ground truth' is known, so that the degree an algorithm discovers that truth is a measure of its quality. The worry then is the authors analysis inserts some variability in a particular way and the primary principle component just 'discovers', the variability that was inserted. I do not know how to definitively test for this, but to check for the obvious things, can the authors:i. Intentionally augment the outer boundary of the wing, and analyze as in Figure 5. Is R >> σ. What if some ripple was added to the boundary, what would that look like mapped to the disk?ii. The authors need to rely on a reference feature to map to the origin of the disk. Assume the position of that was randomized by a few pixels, how would that display in Figure 5?iii. Same set of questions for the way the authors separate the wing from the hinge

We thank the reviewer for identifying these additional checks of the robustness and accuracy of our method. We have included an analysis of all three suggested artificial perturbations of the system in our SI. While the quantitative details of the resulting dependence of σ/R vs R are altered, the qualitative features remain invariant. Crucially, R>>σ for perturbations of reasonable amplitude – see Figure 5—figure supplement 2. In particular, we jiggle the location of hinge cut points and the origin by 3 pixel in each the x and y directions (f and g), and introduce noise in the segmented boundary with the same amplitude.

2. In Figure 2d most of the variation is in common for male-females under the landmark free methods while with landmarks the largest PCA component discovers sex. The authors claim this is a vindication of their method, but could it in fact be a problem (as suggested in 1.)

We include a more detailed description of the relative orientations of the long axis in the data and the axis corresponding to sexual dimorphism in the high-quality images of environmental variation. We observe that within stressed populations of flies, at low diet and high temperatures close to their physiological limit (29°C), the sexual dimorphism axis rotates from being orthogonal to the long axis in the data to parallel – see Figure 2—figure supplement 3. This conditional GxE interaction (G being sex-specific genes) provides further evidence that what we are observing is biological in origin, rather than an artifact of our pipeline. It is also consistent with our observation that alignment of mutational perturbations in signaling genes with the long axis in the data is sometimes dependent on the sex of the flies, depending on the mutation (Figure 5 in manuscript)

Also the data in 2d seems to be bipartite, any explanation?

The bipartite nature of the data at 25C high diet isn’t observed at other conditions – See figure 3 in this response document. All of this data is now included in an additional section of the SI.

3. Can the authors take their conformal mapped and registered images, and extract the coordinates of the Procrustes reference points. What is the relation of the principle components based on the Procrustes variables versus their pixel based measures. It's not entirely clear to me what feature their dominant PC picks up. If I shift a vein normal to itself a lot of pixels change, but the movement is less consequential viewed as an operation on a vein. Thus as usual an information theory measure can be misleading.

We thank the reviewer for identifying two issues in this recommendation. First, precisely for the reasons identified by the reviewer, we take the radon transform of the disc mapped and aligned images of wings. The reason for doing so is so that the anticipated correlated deformation of veins across the ensemble of wings is represented more suitably in radon space. All analysis is performed in the radon space, which as the reviewer is no doubt aware, is an invertible transform. We have now further emphasized this aspect of our analysis in the manuscript.

Second, following the reviewer’s recommendation we present a comparative analysis of the mapped locations of landmarks in the disc and the pixel-based measure. We indeed agree that the landmark centric view of the wing “focuses” on gross deformations of veins, while the pixel-based approach has no knowledge of these macroscopic structures. We thus wanted to assess what the relation between disc mapped landmarks and the dominant direction (PC1) of the pixel-based approach was. In Figure 4C, we present the cloud of landmark data for the 25°C high diet ensemble of wing images. In Figure 4D we present a quantitative analysis of the degree of correlation between the principal components of disc-mapped/aligned landmarks and the pixel-based approach. Most crucially, we observe that PC1 of the pixel-based approach is not captured in the landmark spectrum. Indeed, what we observe is that PC2 is the sexual dimorphism axis and that this is observable in the landmark based approach. Thus, it seems like the variation encoded in PC1 of the pixel-based approaches is not encoded in the landmarking based approach.

4. A key finding is that genetic perturbations fall along the same PC as does intrapopulation variation. In the case of the gene tkv the authors remark that their analysis picks up the same feature displacement in the weak heterozygote as in the homozygote. Is the same true for the other mutants they examine, in particular N. The N mutants are readily visible, if they are processed as in Figure 5b,c would they fall along the primary PC? If the authors consider it too onerous to measure on new flies, can they just by photoshop impose a comparable mutation on a few of their current images and process?

As the reviewer notes, the Notch pathway is important to wing development. One of the mutants we analyzed in heterozygous state is in the mastermind (mam) gene. Mam is a *Drosophila* transcriptional coactivator that specifically functions in the Notch pathway. Mam physically binds to Su(H) in the nucleus but only when Su(H) is complexed to the Notch ICD. Mam recruits p300/CBP to Notch-dependent gene enhancers where they elicit nucleosome acetylation and transcriptional activation. In the wing, mam mutants phenocopy N mutants’ effects on vein patterning. Thus, our analysis of mam heterozygous mutant wings is our proxy for a weak effect N mutant. As the reviewer can see in Figure 5B, the mam data follows the general geometric trend observed in the data as a whole.

5. With their environmental perturbations the authors could ascertain when in development the wings are sensitive to the environment, ie does a molt reset the phenotype of the wing?

This is precisely one of the experiments we are currently pursuing. Figuring out what developmental stage or process buffers the system from perturbations is crucial.

6. The ultimate test of the unique principle component, is to measure the wings on a closely related species of Drosophila (as the authors remark at the end of their discussion). Would this be a big increment in work over what has already been done? Just 10's of individuals from a stock collection of D. simulans, D. schellia or whatever is easiest. Some of these species make sterile hybrids with D. melanogaster, so it would be interesting to know how close the wings are.

Again, the reviewer has predicted what experiments we are currently pursuing. We have indeed acquired the three closely related *Drosophila* species to *D. melanogaster* (all part of the Madagascar radiation 3 Mya) and measurements of wing phenotypes within and across these ensembles are underway. We agree with the reviewer that this is the ultimate test. Additionally, the reviewer is also correct that the patterns of which species can cross is complex, and we intend to study its correlation with phenotypic distance of wings. Again, we thank the reviewer for their prescient insights.

7. Some technical remarks about the mapping to the diski. Is there a reflection among the rotations to align the right and left wings?

First, we performed an analysis to validate the lack of any left-right patterning in the wing, which is what we anticipated based on 100 years of research. An individual’s left and right wings are of course slightly different from each other, but there is no systematic trend across individuals that we observed. Thus, we batch together left and right wings to increase our numbers. This is achieved through reflection. We have now included a description of this in the manuscript.

ii. Is it obvious one should do PCA directly on the pixel variables, (ie not log transformed for instance). Not many pixels change, but those that do may change their gray scale by a factor of 2-10 (depending on how you are treating background). Does this matter?

The reviewer is correct that few pixels change across images, and those that change do so by a significant factor. For example, motion of a vein across a database corresponds to a significant variation in intensity in limited zones of the image. We had previously performed the analysis on log-transformed data, and log-transform of the inverse of data (where veins are bright as opposed to dark structures) to investigate the very issue that is being raised here. We confirmed that all the statistical trends are maintained with intensity transformed data. We now include this analysis in Figure 5—figure supplement 3. Again, we thank the referee for this and other recommendations through which we believe we have provided further evidence towards the accuracy and robustness of the computational approach/pipeline.